# Synthesis of New Zirconium Magnetic Nanocomposite as a Bioactive Agent and Green Catalyst in the Four-Component Synthesis of a Novel Multi-Ring Compound Containing Pyrazole Derivatives

**DOI:** 10.3390/nano12244468

**Published:** 2022-12-16

**Authors:** Mohammed Asiri, Ahmed Ghalib Abdulsalam, Mustafa Kahtan, Fahad Alsaikhan, Issa Farhan, Dhameer A. Mutlak, Salema K. Hadrawi, Muath Suliman, Ritamaria Di Lorenzo, Sonia Laneri

**Affiliations:** 1Department of Clinical Laboratory Sciences, College of Applied Medical Sciences, King Khalid University, Abha 61421, Saudi Arabia; 2Department of Pharmacy, Al-Noor University College, Bartella 46476, Iraq; 3Medical Technical College, Al-Farahidi University, Baghdad 10011, Iraq; 4College of Pharmacy, Prince Sattam Bin Abdulaziz University, Alkharj 11942, Saudi Arabia; 5Medical Physics Department, Al-Mustaqbal University College, Babylon 51001, Iraq; 6AL-Nisour University College, Baghdad 10001, Iraq; 7Refrigeration and Air-Conditioning Technical Engineering Department, College of Technical Engineering, The Islamic University, Najaf 54001, Iraq; 8Department of Pharmacy, School of Medicine and Surgery, University of Naples Federico II, Via D. Montesano, 49-80131 Naples, Italy

**Keywords:** zirconium magnetic nanocomposite, bioactive agent, green catalyst, four-component reaction, pyrazole derivatives, antibacterial activity, antifungal activity

## Abstract

New nanocomposites containing zirconium were synthesized using microwave irradiation. Their structure was confirmed by vibrating sample magnetometer (VSM) curves, X-ray diffraction (XRD) patterns, scanning electron microscope (SEM) and transmission electron microscopy (TEM) images, Fourier transform infrared spectroscopy (FT-IR), and Brunauer–Emmett–Teller (BET) N_2_ adsorption/desorption isotherms. After the structure confirmation of the zirconium magnetic nanocomposite, the catalytic properties in the synthesis of pyrazole derivatives were investigated. Next, the biological activities of the zirconium magnetic nanocomposite, such as the antibacterial and antifungal activities, were investigated. The research results showed that the zirconium magnetic nanocomposite has high catalytic properties and can be used as a magnetic nanocatalyst for synthesizing heterocyclic compounds such as pyrazole derivatives in addition to having high biological properties. The unique properties of the nanoparticles can be attributed to their synthesis method and microwave radiation.

## 1. Introduction

Cyclic organic compounds withwith at least one heteroatom, such as nitrogen, sulfur, and phosphorus, are called heterocycles. Heterocycles have many biological properties. There are heterocyclic compounds in the structures of many drugs. So far, biological properties such as antibacterial, antifungal, anticancer, and antioxidant properties of heterocyclic compounds containing nitrogen, sulfur, and phosphorus have been reported [1,2,3,4,5,6,7]. One of criticalheterocyclic compounds’ critical applications is their use as ligands in complexes. The use of heterocycles as ligands makes the final product retain the biological properties of the heterocycle and the metal, thus having high biological properties. There have been reports of using heterocycles as ligands and synthesizing novel complexes containing various metals such as Cr, Mo, W, gold, and silver. These have unique properties, including anticancer properties [8,9,10,11,12].

The heterocyclic compound of pyrazole with two nitrogen atoms in its structure has many biological properties. This heterocyclic ring is found abundantly in nature, and its valuable derivatives with biological properties have been extracted from natural compounds [13]. Biological properties, such as anticancer [14,15], anti-diabetic [16], antimicrobial [17,18], and antioxidant [19] of heterocyclic compounds containing pyrazole, have been reported. Considering their importance, it is necessary to provide new methods for its synthesis.

Recently, developments in green chemistry and multi-component reactions (MCRs) in synthesizing heterocyclic compounds, particularly for drugs, have increased dramatically. The factors driving this increase include the prevention of environmental pollution, economic efficiency, and high efficiency [20,21]. Another advantage of MCRs is that they perform the reaction in one step, reducing the products’ synthesis time. In multi-component reactions, choosing a suitable catalyst is very important, and recently, nanocatalysts have proven to be effective in these reactions. Nanocatalysts have the added benefit of being recyclable and compatible with green chemistry. [22]. There have been several reports of various nanocatalysts, including metal oxide nanoparticles [23,24,25], magnetic nanoparticles [2,26], supported nanoparticles [27], and metal-organic framework nanoparticles [28], in the synthesis of organic and heterocyclic compounds.

Metal-organic framework nanoparticles are compounds with unique capabilities. High specific surface area and porosity are the main capabilities of these nanostructures, which have resulted in their application as flexible supercapacitors [29] and electrochemical biosensors for biomedical analysis [30,31]. There have been reports of these compounds being efficient and recyclable catalysts in synthesizing organic and heterocyclic compounds [32,33].

This study synthesized a new zirconium magnetic nanocomposite using Fe_3_O_4_ nanoparticles, dipicolinic acid, ZrCl_4_, and a microwave-assisted method. The final product’s structure was identified and confirmed using VSM, XRD, SEM and TEM images, FT-IR, and BET. The final product was used as a catalyst candidate to synthesize new pyrazole derivatives and antimicrobial agents.

## 2. Materials and Methods

### 2.1. Solvents and Raw Materials

The high-purity solvents and raw materials used in this study were purchased from Merck and Sigma-Aldrich. The Fe_3_O_4_ nanostructures were prepared from Sigma-Aldrich. No purification of the raw materials was carried out (Merck KGaA, St. Louis, MO, USA).

### 2.2. Zirconium Magnetic Nanocomposite Synthesis

For the zirconium magnetic nanocomposite synthesis, Fe_3_O_4_ nanoparticles (2 mmol), dipicolinic acid (4 mmol), and ZrCl_4_ (2 mmol) were added to 30 mL double-distilled water and stirred at 80 °C. After 10 min, the solution was put into a microwave and irradiated at a microwave power of 450 W at room temperature. After 10 min, the mixture cooled (room temperature), and the desired product was isolated using an external magnet. The synthesized zirconium magnetic nanocomposite was washed several times with a mix of double-distilled water and ethanol and dried at an ambient temperature.

### 2.3. Synthesis of 1,4-Dihydropyrano[2,3-c]pyrazole-5-Carbonitrile Derivatives Using Zirconium Magnetic Nanocomposite as a Catalyst

For the synthesis of 1,4-dihydropyrano[2,3-c]pyrazole-5-carbonitrile derivatives, zirconium magnetic nanocomposite (4 mg), aromatic aldehydes (1 mmol), and malononitrile (1 mmol) were added to 2 mL EtOH:H_2_O and stirred at 50 °C for 5 min. Then, phenylhydrazine (1 mmol) and ethyl acetoacetate (1 mmol) was added and stirred at 50 °C. The reaction was monitored by thin-layer chromatography. After the completion of the reaction, the zirconium magnetic nanocomposite was separated using an external magnet. The synthesized 1,4-dihydropyrano[2,3-c]pyrazole-5-carbonitrile derivatives were purified using recrystallization in ethanol.

After separation by the magnet, the zirconium magnetic nanocomposite was washed several times with a mixture of double-distilled water and ethanol and was reused after drying at room temperature.

***6-amino-3-methyl-1,4-diphenyl-1,4-dihydropyrano[2,3-c]pyrazole-5-carbonitrile*** (**5a**) FT-IR (KBr, cm^−1^): 3402, 3284, 3181, 2203, 1675, 1660, 1529, 1461, 1317, 1240, 1101. ^1^H-NMR (250 MHz, DMSO-d6): δ (ppm) 7.64 (s, 1 H), 7.41–7.27 (m, 11 H), 4.73 (s, 1 H), 1.63 (s, 3 H). ^13^C-NMR (75 MHz, DMSO-d6): δ (ppm) 159.24, 144.86, 143.92, 143.61, 137.01, 128.92, 128.52, 128.07, 127.69, 127.12, 126.34, 119.96, 112.51, 97.96, 58.13, 11.73. 

***6-amino-4-(4-methoxyphenyl)-3-methyl-1-phenyl-1,4-dihydropyrano[2,3-c]pyrazole-5-carbonitrile*** (**5c**) FT-IR (KBr, cm^−1^): 3392, 3342, 3112, 2165, 1673, 1525, 1416, 1341, 12,120, 1114. ^1^H-NMR (250 MHz, DMSO-d6): δ (ppm) 7.69 (d, 2 H, J = 8.4 Hz), 7.42 (t, 2 H, J = 8 Hz), 7.35 (t, 1 H, J = 7.8 Hz), 7.19 (d, 4 H, J = 8.4 Hz), 6.95 (s, 2 H), 4.63 (s, 1 H), 3.62 (s, 3 H), 1.82 (s, 3 H). ^13^C-NMR (75 MHz, DMSO-d6): δ (ppm) 160.82, 145.62, 144.01, 143.25, 137.82, 129.07, 128.15, 128.52, 127.67, 127.35, 126.73, 119.03, 112.46, 97.34, 57.57, 12.01.

***6-amino-4-(4-hydroxyphenyl)-3-methyl-1-phenyl-1,4-dihydropyrano[2,3-c]pyrazole-5-carbonitrile*** (**5i**) FT-IR (KBr, cm^−1^): 3381, 3313, 3157, 2189, 1670, 1538, 1402, 1321, 1249, 1157. ^1^H-NMR (250 MHz, DMSO-d6): δ (ppm) 9.28 (s, 1 H), 7.71 (d, 2 H, J = 8.4 Hz), 7.55–7.49 (t, 2 H, J = 8.7 Hz), 7.29–7.21 (t, 1 H, J = 8.4 Hz), 7.04 (s, 2 H), 7.11–7.14 (d, 2 H, J = 7.5 Hz), 6.71 (s, 2 H), 4.54 (s, 1 H), 1.77 (s, 3 H). ^13^C-NMR (75 MHz, DMSO-d6): δ (ppm); 160.11, 156.01, 144.98, 143.75, 143.08, 137.21, 133.62, 129.04, 128.66, 126.17, 120.49, 119.34, 114.99, 99.18, 58.61, 12.64.

### 2.4. Zirconium Magnetic Nanocomposite Antimicrobial Activity

To measure the MIC, MBC, and MFC, a concentration of 1–2048 mg/mL of zirconium magnetic nanocomposite and the drug were prepared. The Clinical and Laboratory Standards Institute (CLSI) guidelines (M07-A9, M26-A, M27-A2) were used for the zirconium magnetic nanocomposite antimicrobial activity. Based on the reported methods, relevant tests on the desired Gram-positive, Gram-negative species, and desired fungal species were performed [34,35,36].

## 3. Results

### 3.1. Results of Synthesis and Confirmation Structure of Zirconium Magnetic Nanocomposite

A new zirconium magnetic nanocomposite using Fe_3_O_4_ nanoparticles, pyridine-2,6 dicarboxylic acid, and zirconium (IV) chloride was synthesized under microwave irradiation. Various techniques and analyses, such as vibrating sample magnetometer curves (VSM, Magnetic Daghigh Danesh Pajoh Co, Iran, Kashan), X-ray diffraction patterns (XRD, Philips XPERT PRO, Netherlands, Eindhoven), scanning electron microscope images (SEM, Hitachi S-4800 FESEM, Japan, Tokyo) and transmission electron microscopy images (TEM, Philips EM 208S, Netherlands, Eindhoven), Fourier transform infrared spectroscopy (FT-IR, Thermo Scientific Nicolet-6700, Waltham, MA, USA), and Brunauer–Emmett–Teller N_2_ adsorption/desorption isotherms (BET, Micromeritics, TriStar II 3020 analyser, Norcross, GA, USA), were used to identify and confirm the structure of the zirconium magnetic nanocomposite.

The magnetic saturation of the zirconium magnetic nanocomposite, as shown in the VSM curve in Figure 1, was 0.014 emu/g.

The magnetic property of the zirconium magnetic nanocomposite was compared with the magnetic property of the Fe_3_O_4_ nanoparticles. According to previous reports, the saturation value of the Fe_3_O_4_ nanoparticles was 0.055 emu/g [2]. The decrease in the magnetic saturation of the zirconium magnetic nanocomposite shows that the Fe_3_O_4_ nanoparticles were covered in groups.

The XRD pattern of the zirconium magnetic nanocomposite (Figure 2) confirmed the crystalline structure and the presence of Fe_3_O_4_ nanoparticles in the final product’s structure [37].

The SEM and TEM images of the zirconium magnetic nanocomposite (Figure 3) confirmed the uniformity of the structure and the morphology of the final product. In addition, the SEM and TEM images proved that the structure of the compound was in the nano-sized range.

The FT-IR spectrum of the zirconium magnetic nanocomposite, as shown in Figure 4, proved the desired absorptions of the final product’s structure.

The Fe_3_O_4_ nanoparticles showed absorption in the region of 526 cm^−1^. The peaks in areas 617 cm^−1^ and 667 cm^−1^ were related to Zr-O [38]. The peak observed in the region 1124 cm^−1^ corresponded to the C-O group. The C=C and C=O groups showed absorption in areas 1548 cm^−1^ and 1651 cm^−1^, respectively. The absorption of the C-H group was observed in areas 2930 cm^−1^ and 2970 cm^−1^. Finally, the absorption of the O-H group was observed in the 3400 cm^−1^.

The N_2_ adsorption/desorption isotherms of the zirconium magnetic nanocomposite were the fourth type of the classical isotherm series [39]. The specific surface area of the zirconium magnetic nanocomposite was about 1850 m^2^/g (Figure 5).

As an overall finding, it can be stated that the synthetic compound had magnetic properties and could be easily separated in catalytic reactions. The desired elements and functional groups were observed in the structure of the synthesized compound. The synthesis method and microwave radiation caused uniform morphology and nano-sized particles. In addition, the synthesis method increased the specific surface area, which led to its use as an efficient catalyst and bioactive agent.

Based on the observations and spectral analysis, the following structure was suggested for the zirconium magnetic nanocomposite (Figure 6).

### 3.2. Results of Four-Component Synthesis of Multi-Ring Compound Containing Pyrazole Using Zirconium Magnetic Nanocomposite

From the four-component reaction of the aromatic aldehyde derivatives, malononitrile, phenylhydrazine, and ethyl acetoacetate in the presence of the zirconium magnetic nanocomposite as a catalyst, 1,4-dihydropyrano[2,3-c]pyrazole-5-carbonitrile derivatives were synthesized (Figure 1).

To synthesize the derivatives, we first optimized the reaction conditions, such as the solvent, amount of catalyst, and temperature. Different solvents, such as EtOH, EtOH: H2O (1:1), MeOH, and CH3CN, were tested during optimization. Based on the obtained results, the highest efficiency was obtained using EtOH: H_2_O.

To optimize the catalyst amount, the reactions in amounts of 1–5 mg were tested. Based on the obtained results, high efficiency was observed in using 4 mg as the catalyst. Finally, temperature optimization was performed, and the reaction at 50 °C had the highest yield. The optimization results are given in Table 1.

The structures of the 16 derivatives of 1,4-dihydropyrano[2,3-c]pyrazole-5-carbonitrile (5a–o) synthesized in this study under optimal conditions are given in Table 2.

For synthesizing the 1,4-dihydropyrano[2,3-c]pyrazole-5-carbonitrile derivatives studied in this research using the zirconium magnetic nanocomposite as a catalyst, the Figure 2 mechanism was proposed.

Table 3 shows a comparison of the recently reported synthesis methods of 1,4-dihydropyrano[2,3-c]pyrazole-5-carbonitrile derivatives such as triazine-based functionalized HY zeolite [40], SBA-15/hydrotalcite/heteropoly acid (phosphotungstic acid) [50], tungstic acid immobilized on zirconium-L-aspartate amino acid metal-organic framework-grafted L-(+)-tartaric acid-stabilized magnetic Fe_3_O_4_ nanoparticles [51], yttrium iron garnet [52], triphenylphosphine [53], sugarcane bagasse ash-based silica-supported boric acid [54], and 1,3-dimethyl-2-oxo-1,3-bis(4-sulfobutyl) imidazolidine-1,3-diium hydrogen sulfate[DMDBSI]2HSO4 [55] as a catalyst.

As a result, the zirconium magnetic nanocomposite synthesized the desired product with better results, including higher efficiency, a shorter time, and a lower temperature.

As mentioned earlier, the favorable conditions of the zirconium magnetic nanocomposite as a catalyst can be attributed to its high specific surface area.

Another essential advantage of the catalyst studied in this study was its recycling ability.

Figure 7 shows that the zirconium magnetic nanocomposite can be reused up to six times (for 5a), which does not significantly reduce the efficiency of the product.

### 3.3. Results of Biological Activity of Zirconium Magnetic Nanocomposite

The high specific surface area of the zirconium magnetic nanocomposite synthesized in this study significantly affected the Gram-positive, Gram-negative, and fungal species. Zirconium magnetic nanocomposite’s antimicrobial effects were tested based on the MIC (minimum inhibitory concentration), MBC (minimum bactericidal concentration), and MFC (minimum fungicidal concentration) parameters (Table 4).

The antibacterial effects of nanoparticles were examined on *Staphylococcus epidermidis* and *Bacillus cereus* (Gram-positive), *Klebsiella pneumonia* and *Shigella dysenteriae* (Gram-negative), and *Candida albicans* (fungi).

Furthermore, the antimicrobial effects of some commercial drugs (Cefazolin as an antibacterial drug and Terbinafine as an antifungal drug) on the studied species were tested to compare their effectiveness to the zirconium magnetic nanocomposite.

The results of the antimicrobial tests proved that the zirconium magnetic nanocomposite positively affected all studied Gram-positive, Gram-negative, and fungi species. The MBC value was 16 μg/mL on *Staphylococcus epidermidis*, 128 μg/mL on *Bacillus cereus*, 64 μg/mL on *Klebsiella pneumonia*, and 128 μg/mL on *Shigella dysenteriae*, and the MFC value on *Candida albicans* was 128 μg/mL.

It is noteworthy that Cefazolin was ineffective on Bacillus cereus and Shigella dysenteriae, and Terbinafine was ineffective on *Candida albicans*. However, the zirconium magnetic nanocomposite had a positive effect.

As mentioned earlier, the unique properties of the zirconium magnetic nanocomposite can be attributed to its high specific surface area, which is the result of its synthesis method.

## 4. Conclusions

In the present study, a zirconium magnetic nanocomposite was synthesized using the microwave method. Analyses such as vibrating sample magnetometer curves, X-ray diffraction patterns, scanning electron microscope and transmission electron microscopy images, Fourier transform infrared spectroscopy, and Brunauer–Emmett–Teller N_2_ adsorption/desorption isotherms to identify and confirm its structure were performed. The results of the analyses showed that the synthesis method caused uniform morphology and increased the specific surface area of the zirconium magnetic nanocomposite. The synthesized zirconium magnetic nanocomposite was used as a catalyst in the synthesis of 1,4-dihydropyrano[2,3-c]pyrazole-5-carbonitrile derivatives due to its unique properties, including its high specific surface area, which is essential for catalytic applications. The catalytic activity results compared to the previously reported methods for synthesizing 1,4-dihydropyrano[2,3-c]pyrazole-5-carbonitrile derivatives were significant. One of the other advantages of its use as a catalyst is its possible reuse without a noticeable decrease in efficiency. The high specific surface area of the zirconium magnetic nanocomposite resulted in biological activity, which was effective on Gram-positive, Gram-negative, and the studied fungal species. The noteworthy finding of the antibacterial activity was its higher effectiveness compared to the commercially used drugs.

## Data Availability

The authors confirm that the data supporting the findings of this study are available within the article.

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
