# Peer review of "Synthesis of New Zirconium Magnetic Nanocomposite as a Bioactive Agent and Green Catalyst in the Four-Component Synthesis of a Novel Multi-Ring Compound Containing Pyrazole Derivatives"

_nanomaterials, 2022, doi:10.3390/nano12244468_

Round 1
Reviewer 1 Report
In the manuscript the authors deal with multicomponent one-pot reactions for a sustainable access to biologically relevant heterocyclic systems, e. g. pyrazoles.
Nanocatalysts, i. e. zirconium magnetic nanocomposites, are used due to easy separation, recovery, and reuse.
Synthesis of nanoparticles is straightforward using standard microwave techniques. The magnetic behavior, crystallinity, SEM and TEM, BET adsorption, as well as FT-IR serve for structural prove of the nanocomposites. Based on these results a potential structure was proposed. This structural model is obvious but final prove is not given (and – in my opinion - not necessary for the following study).
The nanocomposites are then used as catalyst in a four-component one-pot synthesis of pyrazole derivatives. Solvent optimization identified EtOH:water (1:1) mixture as best medium. On a small scale (1-5 mg), scope and limitation of the reaction was tested. Reactions occur fast (<1 h) with usual yields > 85% and a high recyclability; a comparison with selected similar synthetic strategies published in literature show the benefilts of the new catalyst.
Somewhat unrelated to the synthetic results, just to exploit the high surface area, the biological activity of the zirconium magnetic nanocomposites against Gram-positive and – negative bacteria as well as fungi were tested.
The reported results are interesting and of synthetic value. For a synthetically interested chemist, the “anti-bacterial results” are probably less interesting. If the focus of the paper is on the characterization of a new nanocomposites and potential applications in general, the biological activity can be part of the manuscript. However, in the introduction the focus is on synthesis, the discussion of antimicrobial effect in line 60-67 has the character of a conclusion/summary of the reported results. Therefore, the biological study seems to be somewhat out of place.
Author Response
Response: Thanks for your attention
Authors would like to thank you for the very insightful and constructive comments and suggestions and your opinions increased the quality of our manuscript. The proposed amendment was made.
Reviewer 2 Report
This manuscript descibes a novel synthesis of nanocomposite containing Zr and Fe. The related nanocomposite was identified by XRD, SEM, TEM and IR. The important applications for new nanocomposite in the one-pot four-component synthesis of dihydropyrano[2,3-c]pyrazole compounds, and related biological activities were investigated. The corresponding applications show the importance of particles. I recommend to accept the manuscript for publication in Nanomaterials.
Author Response
Response: Thanks for your attention
Reviewer 3 Report
Synthesis of new zirconium magnetic nanocomposite as a bioactive agent and green catalyst in the four component synthesis of novel multi-ring compound containing pyrazole derivatives.
This manuscript focuses on the synthesis of a series of nanocomposites containing zirconium using microwave irradiation. Their structure was confirmed by Vibrating Sample Magnetometer curves (VSM), X-Ray Diffraction Pattern (XRD), Scanning Electron Microscope (SEM) and Transmission Electron Microscopy (TEM) images, Fourier Transform Infrared spectroscopy (FT-IR) and Brunauer Emmett Teller N2 adsorption/desorption isotherm (BET). The catalytic properties in the synthesis of pyrazole derivatives were investigated. The biological activities of zirconium magnetic nanocomposite, such as antibacterial and antifungal, were investigated. The research results showed that zirconium magnetic nanocomposite, has high catalytic properties and can be used as a magnetic nanocatalyst in synthesizing of the heterocyclic compounds such as pyrazole derivatives. The pyrazole derivatives and the zirconium magnetic nanocomposite, have high biological properties (antibacterial and antifugal).
The Chapter Results and Discussion is of good scientific quality and the rich and instructive graphic realizes the understanding of the obtained results and of their significance. The experimental data is described appropriately and the manuscript needs no language and grammar corrections.
The manuscript is written straight forward.
The study is a meaningful suppliment to the series of publications regarding the heterocyclic compounds (with P, S, N atoms) related to natural products: synthesis, structural analysis and investigation of their biological activity, that have been extensively studied because their important properties and applications, especially in biological activities, such as, anti-microbial, anti-proliferative (prostate cancer cells), anti-cancer , anti-influenza and with antioxidant activity. In Introduction the autors did not reflect any other field of another heterocyclic with the important applications as chiral ligands for metal catalyst or receptors especially in biological activities.
I believe that you should describe in the introduction more generally to the use of heterocyclic derivatives with biological activity. Therefore, you should cite more special topics. It is of interest for synthetic chemists.
The authors well have also to present in the introduction part the data about the other heterocyclic derivatives, heterocyclic phosphorous compounds with antioxidant and cytostatic activity.
Examples of relevant publications are given below. It is recommended to the authors to cite these papers in their introduction a wider base.
The medicinal chemistry of gold complexes as anticancer drugs, Ott, I. On, Coord. Chem. Rev. 2009, 253, 1670– 1681;
1.3-Dimethyl-l,3-diaza-2-R-5,6-benzo-2 λ3 -phosphorinan-4-ones (R = F, Me2N, 2-Methylpiperidino, MeC(:O)NH-) as Ligands in Transition-Metal Complexes; Synthesis and Structure of Dichloro-Platinum(II)- and Tetracarbonyl Metal(0) Coordination Compounds (Metal = Cr, Mo and W), Plinta H.-J., Neda I.,Schmutzler R., Zeitschrift fur Naturforschung - Section B Journal of Chemical Sciences, 1994, 49,100;
N-Heterocyclic carbenes derived from imidazo-[1,5-a]pyridines related to natural products: synthesis, structure and potential biological activity of some corresponding gold(I) and silver(I) complexes, Mihorianu, M.; Franz, M.H.; Jones, P.G.; Freytag, M.; Kelter, G.; Fiebig, H.H.; Tamm, M.; Neda, I., Appl. Organomet. Chem. 2016, 30, 581–589;
Synthesis, structure, and reactivity of tetrakis(o,o-phosphorus)-bridged calix[4]resorcinols and their derivatives, Vollbrecht A., Neda I.,Thonnessen H., Jones P.G.,Harris R.K., Crowe L.A., Schmutzler R.,Chemische Berichte, 1997, 130, 1715 – 1720.
Some references should be inserted. In conclusion of my review, I recommend this manuscript for publication with minor revisions

Author Response
Response:
Authors would like to thank you for the very insightful and constructive comments and suggestions and your opinions increased the quality of our manuscript. The mentioned items were applied.